# Detection of *mcr-1-1* Positive Enteropathogenic *Escherichia coli* Isolates Associated with Post-Weaning Diarrhoea in an Organic Piglet-Producing Farm in Austria

**DOI:** 10.3390/microorganisms12020244

**Published:** 2024-01-24

**Authors:** René Renzhammer, Lukas Schwarz, Adriana Cabal Rosel, Werner Ruppitsch, Andreas Fuchs, Erwin Simetzberger, Andrea Ladinig, Igor Loncaric

**Affiliations:** 1University Clinic for Swine, Department for Farm Animals and Veterinary Public Health, University of Veterinary Medicine, Veterinärplatz 1, 1210 Vienna, Austria; lukas.schwarz@vetmeduni.ac.at (L.S.); andrea.ladinig@vetmeduni.ac.at (A.L.); 2Austrian Agency for Health and Food Safety, 1090 Vienna, Austria; adriana.cabal-rosel@ages.at (A.C.R.); werner.ruppitsch@ages.at (W.R.); 3VETworks Strengberg, 3314 Strengberg, Austria; strengberg@vetworks.at (A.F.); seitenstetten@vetworks.at (E.S.); 4Institute of Microbiology, Department of Pathobiology, University of Veterinary Medicine, Veterinärplatz 1, 1210 Vienna, Austria; igor.loncaric@vetmeduni.ac.at

**Keywords:** organic farm, *E. coli*, EPEC, colistin resistance, *mcr-1*, whole genome sequencing, post-weaning diarrhoea, PWD

## Abstract

Postweaning diarrhoea (PWD) is a frequent multifactorial disease occurring in swine stocks worldwide. Since pathogenic *Escherichia* (*E*.) *coli* play a pivotal role in the pathogenesis of PWD and porcine *E. coli* are often resistant to different antibiotics, colistin is frequently applied to treat piglets with PWD. However, the application of colistin to livestock has been associated with the emergence of colistin resistance. This case report describes the detection of the colistin resistance gene *mcr-1-1* in two *E. coli* isolated from piglets with PWD in an Austrian organic piglet-producing farm, which was managed by two farmers working as nurses in a hospital. Both *mcr-1*-positive *E. coli* were further analysed by Illumina short-read-sequencing, including assemblies and gene prediction. Both isolates belonged to the same clonal type and were positive for *eaeH* and *espX5,* which are both virulence genes associated with enteropathogenic *E. coli* (EPEC). Due to the detection of *mcr-1*-positive EPEC and based on the results of the antimicrobial resistance testing, the veterinarian decided to apply gentamicin for treatment instead of colistin, leading to improved clinical signs. In addition, after replacing faba beans with whey, PWD was solely observed in 2/10 weaned batches in the consecutive months.

## 1. Introduction

While organic farming systems are perceived by society as having better animal welfare compared to conventional farming systems, restrictions on antibiotic usage and the limited selection of commercially available feed and feed supplements licensed for organic farms can complicate treatment and prophylaxis of diseased piglets. For example, most kinds of feed for organic farms lack animal-based protein sources and rather have large quantities of legumes like faba beans being associated with intestinal inflammation of weaned piglets. Intestinal inflammation of weaned piglets may consequently lead to post-weaning diarrhoea (PWD), which is one of the most frequent diseases in conventional and organic pig herds [1]. In general, PWD is a multifactorial disease of weaned piglets leading to high economic losses due to increased mortality rates, decreased growth rates and increased antibiotic usage [2]. Since the abrupt transition from highly digestible milk to less digestible solid feed containing predominantly complex carbohydrates and plant-based proteins is also associated with post-weaning anorexia and intestinal inflammation, providing a smoother transition by offering creep feed and weaner’s diet with animal-based protein sources can help to ease symptoms of PWD [1]. In addition to predisposing nutritive factors, enterotoxigenic *Escherichia* (*E*.) *coli* (ETEC) play a pivotal role in the pathogenesis of PWD [3]. Besides ETEC, enteropathogenic *E. coli* (EPEC) are also described to cause diarrhoea by intimin-regulated attachment and effacing lesions of enterocytes via the release of effectors encoded by a type III secretion system [3,4]. While previously zinc oxide was predominantly applied successfully to treat piglets with PWD, the usage of therapeutic zinc oxide was banned by the European Commission in 2022 [5]. Therefore, antibiotics are frequently applied to treat piglets with PWD instead [6]. However, as porcine *E. coli* are often resistant to various antibiotic substances, colistin is preferably chosen for the treatment of piglets with PWD [7,8]. Nonetheless, the use of colistin is restricted in livestock as it is used to treat humans with infections caused by multi-resistant Gram-negative bacteria [9]. Since excessive treatment of livestock animals with colistin may lead to the emergence of colistin resistance genes, which can be easily transferred among *Enterobacteriaceae* from different hosts, the usage of colistin to treat livestock animals is currently under debate [10]. Although colistin resistance rates of porcine *E. coli* are considered to be low, case reports on colistin-resistant *Enterobacteriaceae* from swine stocks emphasise the role of pig production in the emergence of colistin-resistant *Enterobacteriaceae* [11,12,13,14]. While the importance of colistin is often underlined in the context of one health, the pivotal role of colistin in the pig production to treat piglets with PWD often gets neglected. Therefore, prevention and treatment against PWD remains a challenge, especially in organic farms due to previously mentioned disadvantageous feeding systems and restriction in antibiotic usage. The current case report aims to highlight the challenges faced by an organic piglet-producing farm dealing with PWD caused by colistin-resistant *E. coli*.

## 2. Case Description

### 2.1. Farm Description

The case farm was an organic, family-owned piglet-producing farm with 35 sows (Large White × German Landrace) operating in a three-week batch farrowing system and managed by two owners, who were also half-time employed as nurses in a local hospital. There were no other swine stocks within a radius of 5 km as the farm was located in the foothills of the Alps in Lower Austria. Gilts were regularly purchased from another organic farm located in Upper Austria and were kept in a separate quarantine unit for six weeks, where they were vaccinated twice against the porcine parvovirus 1 and *Erysipelothrix rhusiopathiae* (Eryseng^®^ Parvo, Laboratorios Hipra, Girona, Spain). All remaining compartments of the stable were located in a single hall with outdoor climate, separated into different compartments. One week prior to farrowing, sows were transferred to individual free farrowing pens with deep straw bedding. Two days after farrowing all piglets received toltrazuril and iron (Forceris^®^, Ceva Santé Animale, Libourne, France) and were routinely vaccinated against the porcine circovirus 2 (PCV2), *Mycoplasma hyopneumoniae* (Porcilis^®^ PCV M Hyo, Intervet International, Boxmeer, The Netherlands) and *Lawsonia intracellularis* (Porcilis^®^ Lawsonia, Intervet International, Boxmeer, The Netherlands) as well as treated with ivermectin (Ivomec^®^, Boehringer Ingelheim, Toulouse, France) on their 21st day of life. Three weeks after farrowing all five lactating sows of one group and their suckling piglets were relocated into a single pen with deep straw bedding for group suckling. While all piglets had access to creep feed containing wheat, barley, soybeans, oats, peas and pumpkin seed meal from their 21st day of life onwards, according to the farmer, creep feed was not ingested by piglets throughout the whole suckling period. All piglets were weaned at six weeks of age and relocated into two nursery pens with a solid concrete floor, which were washed but not disinfected prior to the transfer of piglets. The pens were divided into an outdoor area with two drinkers and an indoor area with straw bedding and two heated microclimate zones (30 °C). Pelleted feed was offered to the weaned piglets via troughs (16 feeding spots per pen) (Table 1).

### 2.2. Clinical History

Due to the ban of zinc oxide in June 2022, the farmers started to treat piglets with PWD with colistin instead. Despite treatment with colistin, between July 2022 and March 2023, PWD was observed predominantly one week after weaning in every batch in at least one quarter of all piglets. Thus, the herd-attending veterinarian decided to submit four rectal swabs from weaned piglets with PWD for microbiological examination and antimicrobial susceptibility testing, as described before [15]. While high numbers of *E. coli* were isolated from rectal swabs of all four animals, haemolytic *E. coli* could only be recovered from animal 2 and animal 4. All isolates displayed a similar resistance pattern after the agar disk diffusion test as they were resistant to tetracycline and doxycycline but susceptible to ampicillin, piperacillin, cefotaxime, ceftazidime, ciprofloxacin, gentamicin, amikacin, tobramycin, chloramphenicol and trimethoprim-sulfamethoxazole. Since the clinical signs of the weaned piglets did not improve after treatment with colistin, all rectal swabs were enriched at 37 °C overnight in BD™ MacConkey Broth (BD, Heidelberg, Germany) and subcultivated on eosin methylene blue (EMB) agar enriched with 3.5 mg/L colistin [16]. After 48 h *E. coli* colonies were observed on EMB agar recovered from swabs of animal 1 and were further investigated by PCR for the detection of mobile colistin resistance genes (*mcr*) 1–5 and using a DNA microarray-based technology (INTER-ARRAY Genotyping Kit CarbaResist, Bad Langensalza, Germany) [17]. The *mcr-1* gene was detected by using both methods (Appendix A).

### 2.3. Farm Visit and Sampling

Since *mcr-1* was solely detected in non-haemolytic *E. coli*, a farm visit was conducted for the collection of further *E. coli* to be tested for *mcr-1* for the exclusion of other causes of PWD and the identification of potential strategies to prevent PWD in the future. During the farm visit, rectal swabs were taken from four seven-week-old piglets with PWD for microbiological investigation (Table 2). In addition, ten faecal samples each were collected from the floor of both nursery pens and were pooled for flotation and PCR for RNA detection of viruses associated with diarrhoea. In addition, two piglets were euthanised to obtain intestines and lymphatic tissue for histologic examination and the detection of other intestinal pathogens (Table 2).

To evaluate the epidemiological situation of colistin-resistant *Enterobacteriaceae* on the farm and find a potential introduction source of *mcr-1*, five environmental samples, including boot sock samples from the nursery pens, faecal samples from gilts in quarantine, faecal samples from house sparrows roosting in the farm compartments, faecal samples from chickens and dust swabs were taken and tested for colistin resistance genes using EMB agar enriched with colistin [16]. *E. coli* could also be recovered from boot sock samples by microbiological examination on EMB agar enriched with colistin and were positive for *mcr-1* by PCR and CARBADETECT array [17].

### 2.4. Whole Genome Sequencing

Furthermore, the *mcr-1*-positive isolate recovered from the rectal swab of animal 1 and the *mcr-1*-positive isolate recovered from the collected boot sock samples were analysed by whole-genome sequencing (WGS) as described before [23,24,25]. A MagAttract HMW (Qiagen, Hilden, Germany) extraction kit was used for Illumina short-read sequencing. Genomic libraries were generated using Nextera XT (Illumina, San Diego, CA, USA). Libraries were 2 × 150 bp sequenced on a NextSeq 2000 device (Illumina, San Diego, CA, USA). To control the quality of the raw reads, FastQC v0.11.9. was used. To remove adapters, trim the last 10 bp of each read and to remove reads with quality scores under 20, Trimmomatic v0.36 was employed. Genome assemblies were generated with SPAdes v3.15.5 and contigs were filtered for a minimum coverage of 5-fold and a minimum length of 200 bp with SeqSphere+ v9.0.3 (Ridom GmbH, Würzburg, Germany).

The clonal relatedness of *E. coli* isolates was determined by a two-locus sequence typing of data from *fumC* and *fimH* using CHTper. The phylotypes were assessed using the Clermont Typing tool [26,27]. In addition, the sequence type was determined by applying the *Escherichia*/*Shigella* data of EnteroBase [28]. For core-genome multilocus sequence typing, (cgMLST) SeqSphere+ software (Ridom, Münster, Germany) was used [29]. ABRicate v1.0.0 was applied for the detection of antimicrobial resistance genes using ResFinder 4.1 and the Virulence Factor Database, as well as virulence genes of *E. coli* using VirulenceFinder 2.0 [30,31]. Plasmids were detected using PlasmidFinder 2.1 [32]. In addition, the probability of predicting the location of a given *bla* resistance gene in *E. coli* was achieved by applying mlplasmids trained on *E. coli*. Both isolates had all characteristics in common. While a total of 115 described genes potentially associated with virulence were detected, neither genes coding for F4 and F18 fimbriae nor ETEC toxin genes could be detected by whole genome sequencing. However, *eaeH* coding for intimin and *espX5* located on the locus of enterocyte effacement (LEE) and coding for an effector of the type III secretion system of EPEC were detected [33]. In addition, *mcr-1* was predicted to be carried on the plasmid of both investigated *E. coli* types and could be further typed as *mcr-1-1*.

### 2.5. Follow Up

Due to the detection of *mcr-1-1*-positive *E. coli*, the veterinarian decided to use gentamicin instead of colistin to treat piglets with PWD, leading to improved clinical signs. In May 2023 the farmer also decided to change the creep feed to a feed containing wheat, soybean meal, wheat bran and pumpkin seeds and to animate piglets for ingestion of creep feed by offering it more frequently. In addition, faba beans and peas were completely withdrawn from the weaner’s diet and replaced with whey protein. From May to November 2023, PWD was solely observed in 2/10 weaned batches, in approximately one quarter of all weaned piglets per batch.

## 3. Discussion

In general, the overall abundance of colistin-resistant *E. coli* in Austrian swine stocks is considered to be low, as so far there has only been one report on an *mcr-1*-positive *E. coli* recovered from the caecum of a slaughtered pig in Carinthia in 2017 [34]. However, the abundance could be higher than expected, since *E. coli* are not tested routinely for colistin resistance due to the lack of clinical break points for colistin [35]. Thus, PCR may be performed instead to detect genes encoding plasmid-mediated colistin resistance genes like *mcr-1* to *mcr-10* [36,37,38,39,40,41,42,43,44,45].

Despite the fact that colistin is a critically important antibiotic for humans, colistin usage in swine stocks could rise in the European Union due to the ban of zinc oxide in June 2022. Consequently, it is possible that a rise of *mcr-1*-positive *E. coli* and other *Enterobacteriaceae* deriving from swine stocks will be observed in the future [46]. In addition, we assume that the awareness of colistin-resistant bacteria will rise amongst veterinarians and farmers since colistin has widely replaced zinc oxide as the substance applied most frequently to treat piglets with PWD [3]. Thus, similar to the case herd, questions of the presence of colistin-resistant *E. coli* could also arise in other cases of treatment failure.

Nevertheless, colistin should not be applied to the case herd anymore to reduce the selection pressure on *E. coli* and consequently, to prevent further spread of *mcr-1*-positive genes via horizontal plasmid transfer to other *Enterobacteriaceae* of pigs or humans [47]. Since both owners were also working in a local hospital, they could be considered a particular hazard for the introduction of *mcr-1*-positive bacteria into the hospital. On the other hand, it cannot be excluded that *mcr-1*-positive bacteria were introduced by the owners, as they were already isolated from patients in hospitals [36]. While there are no data available on the prevalence of *mcr-1*-positive bacteria in Austrian hospitals, Principe et al. reported a prevalence of 0.5% *mcr*-positive *E. coli* recovered from hospitals in nearby Lombardy [48].

Interestingly, *mcr-1* was detected despite the fact that the farmer did not treat piglets with colistin prior to August 2022. Therefore, it is possible that *mcr-1*-positive bacteria were introduced by purchased gilts, other animals, or humans. However, *mcr-1* was neither detected in faeces from gilts in the quarantine units nor in faeces from chickens or house sparrows. Nonetheless, the humans were not tested for *mcr-1* and the sample size could have been too low due to intermittent shedding of *mcr-1*-positive bacteria.

While both *E. coli* isolates tested by WGS were negative for genes associated with ETEC, *eaeH* and *espX5* could be detected. Since *eaeH* is a gene coding for intimin, which is considered to be the typical adhesion factor of EPEC and *espX5* is located on the LEE encoding for the Type III secretion system of EPEC, the *mcr-1*-positive isolate can be categorised as EPEC [33,49]. However, other genes associated with the virulence of EPEC, such as *bfpB*, *escV* or other genes coding for intimin, were not detected by WGS [33]. Therefore, despite the discussion on the detection of genes instead of proteins, it remains difficult to clearly state whether the *mcr-1-1*-positive *E. coli* was pathogenic or not. Nevertheless, our data emphasise that whole genome sequencing is mandatory for sufficient information on the potential pathogenicity of *E. coli* as both isolates were non-haemolytic and genes such as *eaeH* or *espX5* are usually not included in PCR or microarray panels for routine diagnostic procedures [15].

In addition, the transmission of *mcr-1* to other pathogenic *E. coli* via conjugation or plasmid transfer cannot be ruled out [50]. Among all 13 subtypes of *mcr-1*, so far *mcr-1-1* was detected most abundantly in various species, including *Salmonella enterica*, *Citrobacter* spp., *Shigella* spp. and *Klebsiella* spp. [51]. However, since short-read sequencing was performed instead of long-read sequencing like nanopore sequencing, the localisation of *mcr-1-1* on the plasmid can solely be predicted.

Overall, since the clinical signs of piglets improved after treatment with gentamicin but not after treatment with colistin, it is likely that non-haemolytic EPEC were involved in the pathogenesis of PWD in the respective herd.

While applying the effective antibiotic substance helped to ease symptoms of PWD, PWD was still present on the farm until changing the nutritional management. Since clinical signs of PWD became less abundant after adjusting creep feeding and protein sources in the weaner’s diet, it is most likely that PWD was mainly the consequence of offering feed containing faba beans and peas, as well as the fact that neonatal piglets neglected the ingestion of creep feed [52]. To our experience, finding diets with animal-based protein sources licensed for organic farms is extremely tedious in Austria. As solely 2% of all slaughtered pigs derive from organic farms, the economic interests of nutrition companies on producing highly qualitative feed for organic farms may be limited. Therefore, our case clearly emphasises the challenges for organic swine stocks.

Although RNA of rotavirus A and rotavirus C was detected in collected faecal samples and the role of rotaviruses in the pathogenesis of PWD is discussed controversially, there was no evidence of villous atrophy in the euthanised piglets [53]. However, since samples for histologic investigation were taken from two piglets only, a statement on the exclusion of other pathogens associated with diarrhoea in weaned piglets should be performed with caution. In addition, there was no evidence of disease associated with infections with PCV2, *Lawsonia intracellularis*, *Brachyspira* spp. or endoparasites.

## 4. Conclusions

In conclusion, the detection of *mcr-1* in two non-haemolytic *E. coli* with virulence genes associated with EPEC from piglets with PWD demonstrates the potential threat of colistin-resistant *E. coli* for pigs. As colistin is predominantly applied to treat piglets with PWD and *E. coli* are frequently resistant to other antibiotic substances, the identification of alternative strategies to prevent PWD is urgently needed to avoid the potential emergence of colistin-resistant bacteria in swine stocks.

## Figures and Tables

**Table 1 microorganisms-12-00244-t001:** Composition of the feed offered to weaned piglets.

Feed Component	%
Barley	50.0
Sorghum	11.0
Soybeans	10.7
Maize	8.0
Faba beans	6.3
Alfalfa meal	4.7
Peas	3.0
Oats	3.0
Sunflower seeds	2.0
Wheat bran	1.3

**Table 2 microorganisms-12-00244-t002:** Laboratory diagnostic methods for pathogen detection applied in this study.

Pathogen	Specimen	Number of Sampled Animals	Detection Method	Result	Reference
*E. coli*	Rectal swabs	8	Microbiological examination	*E. coli*	[15]
RVA, RVC	Faeces	Pool	RT-PCR	Positive	[18]
TGEV, PEDV	Faeces	Pool	RT-PCR	Negative	[19]
Helminth eggs	Faeces	Pool	Flotation	Negative	[20]
*Lawsonia intracellularis*	Ileal scraping	2	PCR	Negative	[21]
*Brachyspira hyodysenteriae*, *Brachyspira pilosicoli*	Colonic scrapings	2	PCR	Negative	[21]
PCV2	Ileum, Ln. jejunales	2	qPCR	Negative	[22]

RVA: rotavirus A, RVC: rotavirus C, TGEV: transmissible gastroenteritis virus, PEDV: porcine epidemic diarrhoea virus, PCV2: porcine circovirus type 2.

## Data Availability

All data are available at https://www.mdpi.com/ethics, accessed on 27 December 2023.

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
