# Peer review of "Detection of mcr-1-1 Positive Enteropathogenic Escherichia coli Isolates Associated with Post-Weaning Diarrhoea in an Organic Piglet-Producing Farm in Austria"

_microorganisms, 2024, doi:10.3390/microorganisms12020244_

Round 1

Reviewer 1 Report

Comments and Suggestions for Authors

Line 105 -> References in brackets -> are bold typed from reference 15 onwards -> review the previous ones and/or standardize it in whole document;

Lines 105, 106, 114 -> "E. coli" typing -> species names must be typed in italics, please review throughout text;

Lines 117, 120, 176, 180 -> "mcr-1 gene" , "mcr-1" and "mcr-1"-> mcr1 , as in line 139 -> standardize it in whole manuscript;

Lines 144, 145 -> "fumC and fimH" -> italics typing -> review and correct;

Line 152  -> "bla" -> italics typing -> review and correct;

Line 156 -> "eaeH" and "espX5" -> italics typing -> review and correc;

Line 159 -> It is unclear why the designation mcr-1-1. Your explanation is vague. Is there any additional classification of the mcr-1 gene described in the literature that supports this 1-1 subtyping? If yes, please include it in the text. Otherwise, please correct it to mcr-1 throughout the text, including in the title of the manuscript.

Line 194 -> "et al." -> must be typed in italics -> review and correct; 

It would be nice if mcr1 detection results (PCR) could be shown as supplementary material.

Author Response

Reviewer 1

Line 105 -> References in brackets -> are bold typed from reference 15 onwards -> review the previous ones and/or standardize it in whole document;

Thank you for pointing out the inconsistency in the style of the references in the brackets. Therefore, there are no references in bold anymore.

Lines 105, 106, 114 -> "E. coli" typing -> species names must be typed in italics, please review throughout text;

Now E. coli is written in italics in all paragraphs of the manuscript. This was changed in lines 106, 107, 115, 121, 122, 153, 161, 162.

Lines 117, 120, 176, 180 -> "mcr-1 gene" , "mcr-1" and "mcr-1"-> mcr1 , as in line 139 -> standardize it in whole manuscript;

Thank you for pointing out the inconsistency in the style of mcr-1. Now “mcr-1” was written in italics with “-“ throughout the entire manuscript.

Lines 144, 145 -> "fumC and fimH" -> italics typing -> review and correct;

fumC and fimH are now written in italics in line 154.

Line 152  -> "bla" -> italics typing -> review and correct;

bla is now written in italics in line 161.

Line 156 -> "eaeH" and "espX5" -> italics typing -> review and correc;

eaeH and espX5 are now written in italics in line 165.

Line 159 -> It is unclear why the designation mcr-1-1. Your explanation is vague. Is there any additional classification of the mcr-1 gene described in the literature that supports this 1-1 subtyping? If yes, please include it in the text. Otherwise, please correct it to mcr-1 throughout the text, including in the title of the manuscript.

In the sake of consistency, we decided to write mcr-1 instead of mcr-1-1 in lines 195, 198, 199, 205, 206, 208, 209 and 222. In addition, we added a sentence to discuss the mcr1-1 allele which was detected in the respective E. coli in lines 223 – 225.

Among all 13 subtypes of mcr-1, so far mcr-1-1 was detected most abundantly in various species including Salmonella enterica, Citrobacter spp., Shigella spp. and Klebsiella spp. [52].

Line 194 -> "et al." -> must be typed in italics -> review and correct; 

et al. is now written in italics in line 201.

It would be nice if mcr1 detection results (PCR) could be shown as supplementary material.

We followed the recommendation of the reviewer and added a PCR scan of the mcr-1 PCR to the supplementary material (Figure S1 PCR results).

Reviewer 2 Report

Comments and Suggestions for Authors

The manuscript is well written and provides significant new information on the prevalence of colistin resistance in Austrian pig farms.

Points to consider:

Abstract: Correctly summarizes the work that was done. It should be stated, however (line 22 ff), what type of WGS was performed and furthermore, that bioinformatics (assembly, gene prediction) was used to characterize mcr-1 positive E.coli.

Introduction: Gives the necessary background information. There should be some indication on what previous work the current study is based and what the major findings of the current case description are (insert at line 69 ff)

Whole Genome Sequencing (2.4):

I strongly urge the insertion of relevant information on the WGS approach after the first sentence. Information on the sequencing method and some hints to the bioinformatics part are required to understand the basis for the analysis. E.g. an Illumina Sequencing approach might deliver only relatively short contigs and a fragmented view of the whole genome content, whereas Nanopore produced Sequences might even clearly indicate where (on the chromosome or on a plasmid) important genes (ABR or virulence) are located. 

lines 158-159: please give some details why mcr-1-1 was placed on a plasmid, not the chromosome.

Discussion:

lines 176-177: The mere presence of an intact gene, however, implies that it will be used to produce a functional protein and a phenotype under some circumstances. 

I suggest to delete the sentence because it does not provide any important information.

line 210 ff: It should also be noted that the used short read method can be complemented in future analyses by long read sequencing which allows to unambiguously assign important features to the chromosome or the plasmid(s) which potentially can act as vectors for horizontal transmission.

Supplementary Materials:

Raw sequencing data used to assemble contigs should be deposited in a data repository for public access.

Author Response

Reviewer 2:

Points to consider:

Abstract: Correctly summarizes the work that was done. It should be stated, however (line 22 ff), what type of WGS was performed and furthermore, that bioinformatics (assembly, gene prediction) was used to characterize mcr-1 positive E.coli.

 The authors added information to the applied type of WGS to the abstract in lines 23 and 24:

analyzed by Illumina short-read-sequencing including assemblies and gene prediction.

Introduction: Gives the necessary background information. There should be some indication on what previous work the current study is based and what the major findings of the current case description are (insert at line 69 ff)

 Since this is a case report of a farm dealing with post weaning diarrhoea there was no previous work on the current case.

Whole Genome Sequencing (2.4):

I strongly urge the insertion of relevant information on the WGS approach after the first sentence. Information on the sequencing method and some hints to the bioinformatics part are required to understand the basis for the analysis. E.g. an Illumina Sequencing approach might deliver only relatively short contigs and a fragmented view of the whole genome content, whereas Nanopore produced Sequences might even clearly indicate where (on the chromosome or on a plasmid) important genes (ABR or virulence) are located.

The authors thank the reviewer to add information in the applied WGS. Thus, information on the sequencing method and bioinformatics were added to lines 144 – 152.

MagAttract HMW (Qiagen, Hilden, Germany) extraction kit was used for Illumina short-read sequencing. Genomic libraries were generated using Nextera XT (Illumina, San Diego, CA, USA). Libraries were 2 × 150 bp sequenced on a NextSeq 2000 device (Illumina, San Diego, CA, USA).To control the quality of the raw reads FastQC v0.11.9. was used. To remove adapters, trim the last 10 bp of each read and to remove reads with quality scores under 20 trimmomatic v0.36 was employed. Genome assemblies were generated with SPAdes v3.15.5 and contigs were filtered for a minimum coverage of 5-fold and a minimum length of 200 bp with SeqSphere+ v9.0.3 (Ridom GmbH, Würzburg, Germany).

lines 158-159: please give some details why mcr-1-1 was placed on a plasmid, not the chromosome.

We followed the recommendation of the reviewer and changed the wording in lines 167 – 168.

In addition, mcr-1 was predicted to be carried on the plasmid of both investigated E. coli and could be further typed as mcr-1-1.

Discussion:

lines 176-177: The mere presence of an intact gene, however, implies that it will be used to produce a functional protein and a phenotype under some circumstances. I suggest to delete the sentence because it does not provide any important information.

The sentence in Lines 176 – 177 was deleted.  

line 210 ff: It should also be noted that the used short read method can be complemented in future analyses by long read sequencing which allows to unambiguously assign important features to the chromosome or the plasmid(s) which potentially can act as vectors for horizontal transmission.

We totally agree that long-read sequencing is better to make a clear statement on the localization of mcr-1-1. Therefore, we added a sentence to the discussion in lines 225 – 227.

However, since short-read sequencing was performed instead of long-read sequencing like nanopore sequencing, the localization of mcr-1-1 on the plasmid can solely be predicted.

Supplementary Materials:

Raw sequencing data used to assemble contigs should be deposited in a data repository for public access

Raw sequencing data was registered with the BioProject database.

BioProject ID:  PRJNA1064134

After reviewed by the database staff, the project information will be accessible with the following link within a few days:

http://www.ncbi.nlm.nih.gov/bioproject/1064134
